# Therapeutic Drug Monitoring of Biologics in IBD: Essentials for the Surgical Patient

**DOI:** 10.3390/jcm10235642

**Published:** 2021-11-29

**Authors:** Rodrigo Bremer Nones, Phillip R. Fleshner, Natalia Sousa Freitas Queiroz, Adam S. Cheifetz, Antonino Spinelli, Silvio Danese, Laurent Peyrin-Biroulet, Konstantinos Papamichael, Paulo Gustavo Kotze

**Affiliations:** 1Health Sciences Postgraduate Program, School of Medicine, Pontifical Catholic University of Paraná (PUCPR), Curitiba 80215-901, Brazil; robremernones@gmail.com; 2Division of Colon and Rectal Surgery, Cedars-Sinai Medical Center, Los Angeles, CA 90048, USA; pfleshner@aol.com; 3Department of Gastroenterology, University of São Paulo School of Medicine, Sao Paulo 05403-000, Brazil; nataliasfqueiroz@gmail.com; 4Department of Medicine and Division of Gastroenterology, Beth Israel Deaconess Medical Center, Harvard Medical School, Boston, MA 02215, USA; acheifet@bidmc.harvard.edu (A.S.C.); kpapamic@bidmc.harvard.edu (K.P.); 5Division of Colon and Rectal Surgery, IRCCS Humanitas Research Hospital, Via Manzoni 56, 20089 Milan, Italy; antoninospinelli@gmail.com; 6Department of Biomedical Sciences, Humanitas University, Via Rita Levi Montalcini 4, 20090 Milan, Italy; sdanese@hotmail.com; 7IBD Centre, Humanitas Research Hospital, 20089 Milan, Italy; 8Department of Gastroenterology, University Hospital of Nancy, 54500 Nancy, France; peyrinbiroulet@gmail.com; 9IBD Outpatient Clinics, Pontifical Catholic University of Paraná (PUCPR), Curitiba 80215-901, Brazil

**Keywords:** therapeutic drug monitoring, inflammatory bowel disease, surgery, Crohn’s disease, ulcerative colitis, anti-TNF therapy, vedolizumab, ustekinumab

## Abstract

Despite significant development in the pharmacological treatment of inflammatory bowel diseases (IBD) along with the evolution of therapeutic targets and treatment strategies, a significant subset of patients still requires surgery during the course of the disease. As IBD patients are frequently exposed to biologics at the time of abdominal and perianal surgery, it is crucial to identify any potential impact of biological agents in the perioperative period. Even though detectable serum concentrations of biologics do not seem to increase postoperative complications after abdominal procedures in IBD, there is increasing evidence on the role of therapeutic drug monitoring (TDM) in the perioperative setting. This review aims to provide a comprehensive summary of published studies reporting the association of drug concentrations and postoperative outcomes, postoperative recurrence (POR) after an ileocolonic resection for Crohn’s disease (CD), colectomy rates in ulcerative colitis (UC), and perianal fistulizing CD outcomes in patients treated with biologics. Current data suggest that serum concentrations of biologics are not associated with an increased risk in postoperative complications following abdominal procedures in IBD. Moreover, higher concentrations of anti-TNF agents are associated with a reduction in colectomy rates in UC. Finally, higher serum drug concentrations are associated with reduced rates of POR after ileocolonic resections and increased rates of perianal fistula healing in CD. TDM is being increasingly used to guide clinical decision making with favorable outcomes in many clinical scenarios. However, given the lack of high quality data deriving mostly from retrospective studies, the evidence supporting the systematic application of TDM in the perioperative setting is still inconclusive.

## 1. Introduction

Inflammatory bowel disease (IBD) is characterized by a course of chronic and recurrent bowel inflammation and eventually cumulative and irreversible bowel damage [1,2]. In cases where moderate to severe disease activity is present, biological therapy is the cornerstone treatment as it can prevent disease progression [3]. There are several biological agents approved globally to treat both Crohn’s disease (CD) and ulcerative colitis (UC), with different mechanisms of action: tumor-necrosis factor (TNF) inhibitors, anti-integrins, and anti-interleukins [4,5]. 

One of the most challenging decisions in the treatment of patients with IBD is the choice of the biological agent. Although each of the different biologics has the potential to induce remission, one cannot predict individual response. Approximately one-third of patients may have a primary nonresponse to an initial agent. Secondary loss of response, after initial improvement, is also frequent in the management of CD and UC [6,7]. Data on genetic and microbiological signatures and new biomarkers are needed in order to guide this appropriate medication choice [8]. This is a critical aspect for precision medicine in IBD [9].

Another feature of precision medicine in IBD is founded on the pharmacokinetic study of currently approved drugs, also known as therapeutic drug monitoring (TDM). This strategy, based on measurement of serum concentrations and antibodies to a specific agent, assumes that there are specific thresholds for concentration of biologics above which there is increased chance of induction and maintenance of remission [10]. TDM can be used reactively on evidence of therapy failure or proactively with the goal of anticipating and preventing therapeutic failure [11,12]. There are still many controversies regarding TDM in regard to when it should be performed and which range of drug concentrations should be considered adequate for each agent and clinical scenario. 

Currently, the management of IBD is based in a multidisciplinary approach, including medical and surgical options for different disease phenotypes. Despite the approval of new biologics and small molecules, and newer strategies such as earlier treatment and treat-to-target, surgery is still required in a substantial portion of CD and UC patients [13,14,15]. Patients who are refractory to optimal medical therapy and those with disease complications (e.g., dysplasia, perforation, and strictures) comprise the most common indication for surgery in IBD [16,17,18]. Most patients who undergo surgery have been previously exposed to biologic therapy. Thus, it is essential for the surgeon to understand the relationship between biologic agents and surgery, including situations where serum drug concentrations can influence perioperative outcomes [19,20]. The aim of this review is to summarize essential concepts of TDM for IBD surgeons, by discussing the common clinical situations where it can influence pre-, peri-, and postoperative scenarios in CD and UC, specifically examining TDM in postoperative morbidity, CD recurrence, need for colectomy in UC and perianal fistula treatment.

## 2. Tdm and Postoperative Complications in IBD

### 2.1. Anti-TNF Agents

Currently, there is still controversy whether the preoperative use of biologics impacts postoperative outcomes in IBD. Data regarding serum concentrations of biologics in the perioperative period are based on one large multicenter prospective trial (The Postoperative Infection in Inflammatory Bowel Disease—PUCCINI) and few prospective single-center studies [21,22,23].

A large retrospective study from Waterman et al. [24] including 473 CD-related surgical procedures (195 in patients previously treated with anti-TNFs and 278 in matched controls) was the first to evaluate the association between serum biologic drug concentrations and postoperative complications. It found that detectable infliximab concentrations did not increase the rates of postoperative wound infection (*p* = 0.21). However, only 16 UC patients had preoperative levels measured.

A study from Lau et al. was the first prospective study in patients undergoing surgery for IBD with preoperative evaluation of serum concentrations of infliximab [22]. In this study, 123 patients with CD underwent abdominal surgery. Infliximab concentration higher than 3 μg/mL was related to an increased rate of overall complications (Odds ratio (OR) 2.5; *p* = 0.03) and infectious complications (OR 3.0; *p* = 0.03). Overall complications and readmissions rates were significantly higher in patients with drug concentrations higher than 8 μg/mL. In the UC cohort (n = 94), patients with infliximab concentrations > 3 μg/mL compared to those with drug concentrations ≤ 3 μg/mL had similar rates of adverse postoperative outcomes when stratified according to the specific type of surgery. Postoperative morbidity was seen in 31/77 (40%) patients with undetectable concentrations and in 8/17 (41%) patients with detectable infliximab concentrations (*p* = 0.61).

The large multicenter PUCCINI trial [21], prospectively assessed the risk of surgery and biologics, including IBD patients who underwent abdominal operations. Among 955 procedures (ileocolonic resections (n = 410), small bowel or colonic segmental resections (n = 185), and subtotal colectomy with ileostomy (n = 168)), 382 with use of anti-TNFs up to 12 weeks before surgery, the rates of overall infectious complications were similar between patients previously treated with anti-TNFs and controls (20% vs. 19.4%, *p* = 0.801) or detectable anti-TNF drug concentrations (19.7% vs. 19.6%, *p* = 0.985). In the same vein, similar rates of surgical site infections were found in patients with prior anti-TNF therapy exposure (12.4% vs. 11.5%, *p* = 0.692) or detectable drug concentrations (10.3% vs. 12.1%, *p* = 0.513). Both prior anti-TNF exposure and detectable drug concentrations were not significantly associated with the risk of overall infectious complications or surgical site infections. Data regarding serum concentrations of vedolizumab and ustekinumab and their relationship with postoperative outcomes from the same study are eagerly awaited.

A French study also prospectively analyzed the possible influence of serum concentrations of anti-TNFs on postoperative outcomes after ileocolic resection in patients with CD [23]. From the 209 patients initially included, 76 had serum concentrations of infliximab or adalimumab available prior to surgery. Trough concentrations > 1 ug/mL (OR = 0.69, 95% (confidence interval) CI 0.21–2.22) and > 3 ug/mL (OR = 0.95, 95% CI 0.28–2.96) were not related to an increased rate of postoperative complications.

### 2.2. Anti-Integrins and Anti-Interleukins

Regarding vedolizumab, only one study has assessed the impact of preoperative vedolizumab drug concentrations on postoperative outcomes in patients undergoing major abdominal surgery for IBD [25]. Of the 72 patients with preoperative exposure, 38 (53%) patients had detectable (>1.6 μg/mL), and 34 (47%) had undetectable vedolizumab concentrations. In the UC cohort (n = 42), 48% hadundetectable vedolizumab concentration in contrast to 52% who had a detectable one. Postoperative morbidity was comparable between these groups. The CD cohort included 27 patients, of which 48% had undetectable vedolizumab concentrations. Similar to UC, in the CD cohort (n = 27) there were no statistically significant differences in overall complications between patients with (48%) or without (52%) undetectable vedolizumab concentrations. Interestingly, there was a significantly lower incidence of postoperative ileus in CD patients with detectable vedolizumab concentrations compared to patients with undetectable concentrations (*p* < 0.04). Although the association between vedolizumab and postoperative ileus needs to be validated, it may reflect the ability of this specific agent to bind to the integrin α4β7 receptor present on mast cells [26,27].

Similar to vedolizumab, there are limited data on the effect of preoperative ustekinumab concentrations on postoperative surgical outcomes in IBD. The only report, presented at DDW 2021, included 36 patients with IBD. Ustekinumab concentrations were detectable (≥0.9 μg/mL) in 25 (69%) and undetectable in 11 (31%) patients [28]. Among the patients with detectable drug concentrations, the median ustekinumab concentration was 6.4 μg/mL (range 0.9–25). Overall postoperative morbidity (27% vs. 28%, *p* = 0.72), 30-day readmission rate (18% vs. 8%, *p* = 0.57), postoperative ileus (18% vs. 8%, *p* = 0.57), and wound infection (9% vs. 4%, *p* = 0.52) were comparable between the two groups. 

It is clear that our knowledge gap in the evaluation of serum drug concentrations and postoperative outcomes for non-TNF agents is significant. As previously stated, we eagerly await data from the PUCCINI trial, where serum concentrations of both vedolizumab and ustekinumab and their relationship with postoperative outcomes will be analyzed [21]. Table 1 summarizes available data with anti-TNF agents, vedolizumab, and ustekinumab regarding influence of serum drug concentrations on postoperative outcomes.

## 3. Tdm and Postoperative Recurrence in CD

Even though there is appropriate evidence on the role of anti-TNFs in the prevention of endoscopic recurrence after ileocolonic resection, the effect of drug concentrations on recurrence rates has not been adequately explored and available data is scarce. A recent systematic review identified only four studies which assessed infliximab concentrations and endoscopic postoperative recurrence (POR) in CD, with higher concentrations mostly associated with lower POR rates [29].

The PREVENT (Prospective, Multicenter, Randomized, Double-Blind, Placebo-Controlled Trial Comparing Infliximab and Placebo in the Prevention of Recurrence in Crohn’s Disease Patients Undergoing Surgical Resection Who Are at an Increased Risk of Recurrence) trial [30] evaluating 297 patients who had an ileocolonic resection for CD and received either infliximab or placebo showed that among patients who had endoscopic POR, 52.4% had a week 72 undetectable drug concentration; 31.3% had infliximab concentrations 0.1 to 1.85 μg/mL; 18.8%, 1.85 to 4.44 μg/mL; 26.7%, 4.44 to 7.77 μg/mL and 13.3% had drug concentrations higher than 7.77 μg/mL. Patients with positive, negative, or inconclusive antibodies to infliximab (ATI), had an endoscopic POR rate of 64.7% (11/17), 46.7% (7/15), and 30.1% (22/73), respectively. 

A study from Israel showed that lower infliximab trough concentrations and ATI were associated with endoscopic POR [31]. Significantly higher median infliximab concentrations were found in patients with a Rutgeerts’ score of i0 (3.1 [interquartile range (IQR) 0.1–4.1] μg/mL) as compared to those with a score of i4 (0.1 [IQR 0.1–3] μg/mL; *p* = 0.037). When limited to patients naïve to anti-TNF prior to surgery the difference in infliximab concentrations (2.3 [IQR 0.3–3.8] vs. 1.1 [IQR 0.1–3.3] μg/mL, *p* = 0.048) and ATI (7.7% vs. 60%, *p* = 0.044) remained significant. In the same study, the same association was not observed for 41 adalimumab patients.

In a post hoc analysis of a small randomized controlled trial (RCT), Bodini et al. described the correlation between adalimumab concentrations and POR in six patients treated with monotherapy [32]. Serum concentrations were evaluated every 8 weeks for 2 years. Patients with clinical or endoscopic POR compared to those with clinical or endoscopic remission had lower adalimumab concentrations (median (IQR) [7.5 (4.4–9.8) μg/mL vs. 13.9 (8.9–23.6) μg/mL, *p* < 0.01). 

Boivineau et al. [33], presenting results of 19 CD patients on adalimumab after ileocolic resections showed that serum adalimumab concentrations measured 3 months after surgery were higherin patients with normal mucosa (Rutgeerts’ score ≤ i1) versus those with endoscopic POR (Rutgeerts’ score ≥ i2) (7.95 μg/mL vs. 3.25 μg/mL, respectively, *p* = 0.0485). The same study found an inverse correlation between adalimumab concentrations Rutgeerts’ score (*p* = 0.004), and 86% of patients with concentrations less than 4.2 μg/mL had endoscopic POR compared to 15% of patients with concentrations ≥ 4.2 μg/mL (*p* = 0.025). 

A sub-analysis from the POCER (Postoperative Crohn’s Endoscopic Recurrence) trial demonstrated opposite results. In this study, there were 52 patients with serum concentrations of adalimumab measured after ileocolic resection [34]. When combining endoscopic outcomes from 6 and 18 months, patients in endoscopic remission compared to those with POR (Rutgeerts ≥ i2) had similar adalimumab concentrations (9.98 μg/mL vs. 8.43 μg/mL, respectively, *p* = 0.387). There were also no statistically significant differences (*p* = 0.495) when adalimumab concentrations were compared between each different Rutgeerts’ score category (i0 to i4).

Table 2 summarizes data on the application of TDM in POR in CD. Low serum concentrations of anti-TNF agents and immunogenicity seem to be associated with a higher risk of endoscopic POR in patients undergoing an ileocolonic resection for CD. The role of TDM to better optimize not only anti-TNF therapy, but also biologics with a different mechanism of action for preventing and treating POR, should be evaluated in large prospective studies and RCTs.

## 4. Tdm and Colectomy Rates in UC

Recent data demonstrated a possible relation between low serum concentrations of infliximab and the need for colectomy in UC patients. Papamichael et al. assessed the long-term follow-up of 99 UC patients with primary non-response to infliximab [35]. Lower week 2 and week 6 infliximab concentrations at were found in patients who required colectomy (n = 55) as compared to patients with no need for surgery. An infliximab concentration ≤ 16.5 μg/mL at week 2 week 6 was an independent predictor of colectomy. When stratification of infliximab concentrations was performed in quartiles, patients with concentrations in the lower quartiles (<10 μg/mL) had higher rates of colectomy at weeks 2 (70%) and 6 (89%).

Similar data from the Leuven group described the outcome of 285 UC patients with refractory disease on infliximab [36]. Overall, 57/285 (20%) patients needed colectomy during the disease course. Week 14 infliximab concentrations were available in a subset of patients (n = 112). A serum week 14 infliximab concentration greater than 2.5 ug/mL was predictive both for relapse-free survival (*p* < 0.001) and colectomy-free survival (*p* = 0.034).

Acute severe ulcerative colitis (ASUC), refractory to intravenous corticosteroids (CS), is a challenging condition to treat, and colectomy rates remain high regardless of the efficacy of salvage therapies such as cyclosporine and infliximab [37,38]. A substantial portion of patients do not respond to infliximab, possibly due to low drug exposure as a result of increased disease inflammatory burden and high drug clearance and drug fecal loss [39,40,41,42]. 

The relation of colectomy rates in ASUC and the clearance of infliximab was also recently studied. Battat et al. demonstrated that, in 39 patients with ASUC, those with colectomy at 6 months had higher median baseline calculated infliximab clearance compared to those without (0.733 vs. 0.569 L/day, respectively, *p* = 0.005) [41]. A clearance threshold of infliximab of 0.627 L/day identified patients who underwent colectomy with a sensitivity (SN) and specificity (SP) of 80% and 82.8%, respectively (AUC 0.80). These data described that with higher clearance of the drug, consequent lower serum concentrations are associated with greater chance of colectomy [41]. A study from Kevans et al. including 36 patients with steroid-refractory ASUC showed that longer induction infliximab half-life and lower drug clearance were associated with week 14 clinical response and week 54 at CS-free remission [42].

Based on the currently available data, emphasis should be given to studying the role of TDM in ASUC and choosing the optimal infliximab dosing. Table 3 describes the main studies regarding this topic in detail. Due to paucity of data from prospective studies and RCTs, the American Gastroenterological Association makes no recommendation on routine use of intensive versus standard infliximab dosing in hospitalized adult patients with ASUC being treated with infliximab [43]. The prospective multi-center PROTOS (Pharmacokinetics of IFX and TNF Concentrations in Serum, Stool, and Colonic Mucosa in Acute Severe Ulcerative Colitis) study is currently underway aiming to define infliximab pharmacokinetics and guide infliximab dosing strategies in patients with ASUC.

## 5. TDM and Perianal Fistulizing CD

Current data on the use of TDM and perianal fistulizing CD are limited to anti-TNF agents. Data from adult populations are based on cross-sectional studies or retrospective observational cohorts (Table 4). Most studies demonstrate that there is a positive correlation between infliximab and adalimumab concentrations and fistula closure

A post hoc analysis of ACCENT-II RCT (282 patients after induction therapy and 139 patients on maintenance therapy) recently published by Papamichael et al. [48] demonstrated that higher concentrations of IFX at week 14 were independently associated with composite remission defined as complete fistula closure and CRP normalization at week 14 (OR: 2.32; 95% CI: 1.55–3.49; *p* < 0.001) and week 54 (OR: 2.05; 95% CI: 1.10–3.82; *p* = 0.023). IFX concentrations predictive of composite remission were ≥20.2 μg/mL, ≥15 μg/mL and ≥7.2 μg/mL at weeks 2, 6, and 14, respectively. These correlations comprise the only evidence derived from a prospective trial, despite it being a post hoc analysis.

On the pediatric population, two prospective studies were conducted. El-Matary et al. followed 27 patients prospectively and found that an IFX concentration higher than 12.7 μg/mL at week 14 was associated with fistula healing at week 24 (SN 0.62, SP: 0.65) [50]. In addition to this study, Ruemmele et al. randomly assigned 36 patients to receive standard or high doses of ADA. No statistical difference in fistula closure was demonstrated between the different dosing regimens, and ADA levels at weeks 16 and 52 did not correlate with fistula closure in this underpowered study [51].

The studies of TDM and perianal fistulizing CD are associated with some important limitations, mainly related to how the response to treatment was evaluated (with MRI outcomes), as well as for not clearly defining fistula classification (simple versus complex). Prospective data on TDM for perianal fistulizing CD are awaited. The results of the PROACTIVE (Prospective Randomised Controlled Trial of Adults with Perianal Fistulising Crohn’s Disease and Optimised Therapeutic Infliximab Levels) RCT regarding adults with perianal fistulizing CD and proactively optimized infliximab concentrations) [52] is expected to shed more light regarding the role of proactive TDM in this situation.

## 6. Discussion

This review provides a comprehensive assessment of the data on the role of TDM and surgical outcomes in patients with IBD (Figure 1). Although there are limited studies supporting the widespread use of the TDM in the perioperative setting, there is growing evidence demonstrating its potential benefits. 

Abdominal and perianal surgery in IBD patients demands high expertise and a multidisciplinary approach. IBD patients are frequently malnourished, with a high inflammatory burden, often have a past history of previous surgeries, and are frequently exposed to biologics, corticosteroids, and immunomodulators [53]. Consequently, the surgical complications are expected to be higher among these patients than among patients undergoing abdominal surgery for other reasons [54,55]. Thus, early studies identifying the potential relation between postoperative complications and preoperative exposure to biologics should be considered carefully, as numerous confounding factors might influence surgical outcomes in this population. Prospective studies with precise biomarkers, such as quantification of tissue penetration of the drugs in surgical specimens are encouraged to better describe the effect of preoperative biologics on postoperative complications.

Although a positive exposure–response relationship between higher drug concentrations and favorable therapeutic outcomes has been consistently demonstrated [10], this association seems less clear in the context of postoperative recurrence of CD. In the study by Fay et al. [31], despite the significant difference between groups with or without POR, low IFX concentrations (2.4 [0.45–4.1] μg/mL) were still observed in patients with no POR supporting the hypothesis that the actual threshold in the postoperative scenario can be somewhat different than in luminal CD, without a prior surgery. Moreover, it should be emphasized that disease recurrence could be significantly influenced by the higher biologic clearance as a consequence of more severe disease at baseline along with the amount of residual inflamed bowel, a confounding factor poorly explored in the available literature.

There is growing data suggesting the use of proactive TDM during induction for UC and likely for ASUC. Post hoc analyses of the ACT-1 and ACT-2 RCTs demonstrate a positive correlation between infliximab concentrations and favorable outcomes, such as clinical response and remission as well as mucosal healing. Higher infliximab concentrations were also associated with higher rates of week 8 mucosal healing in patients with UC [56]. However, trials evaluating accelerated IFX induction regimen in the setting of ASUC are controversial. A recent meta-analysis of seven studies did not show any difference in in-hospital colectomy rates between accelerated infliximab induction therapy and standard induction therapy [57]. However, there were likely significant confounding factors in these studies. Given that ASUC constitutes a life-threatening condition with reported mortality rates reaching 1% in a recent systematic review and meta-analysis from population-based studies [58], RCT accounting for disease severity and IFX pharmacokinetics are warranted to define the best IFX dosing strategy.

Perianal fistulas are a disabling complication of CD and can have a significant impact on patients’ quality of life [59]. Unfortunately, closure of fistula tracts and radiologic healing are considered ambitious outcomes with low remission rates with any therapy, even anti-TNF. Recent studies regarding anti-TNF therapy have shown that higher than previously reported drug concentrations might be needed to attain complete healing of fistulae [44,45]. It has been suggested that increasing the dose of anti-TNFs with the aim of achieving higher drug concentrations could be helpful in this setting [18]. However, whether this is related to improved mucosal healing rather than a direct action of the drug on fistula tracks remains unclear. Notably, a recent pilot study investigating the role of tissue drug concentrations in fistula tracts of CD patients on anti-TNF therapy found an absence of drug detection in fistula tissue [60]. These observations increase uncertainties surrounding the potential role of tissue penetrance of anti-TNF agents of in response to treatment. 

## 7. Conclusions

The use of TDM is becoming more available globally. Application of serum concentrations and antibody measurement with anti-TNF agents is most commonly used. As indications for surgery in CD and UC persist, despite important advances in medical management of IBD, it is important to define the possible impact of biological agents in the perioperative period. Thus, surgeons should be aware of the possible practical application of the use of TDM in the treatment of IBD and in the peri-surgical period.

Detectable serum concentrations of biologics do not appear to increase postoperative complications after abdominal procedures in IBD. Higher concentrations of anti-TNF agents are associated with a reduction in colectomy rates in UC and may have a role in those admitted with ASUC. Mirroring luminal disease, higher concentrations of anti-TNF agents seem to be associated with reduced rates of postoperative recurrence after ileocolonic resections and higher rates of perianal fistula healing in CD.

Precision medicine is a natural consequence of the development of diagnostic methods and therapeutic agents in IBD. Application of TDM in surgical patients may be an important piece of the “right therapy to the right patient at the right time” aphorism. More prospective data analyzing the relation of serum concentrations of biologics in the perioperative period are awaited.

## Figures and Tables

**Figure 1 jcm-10-05642-f001:**
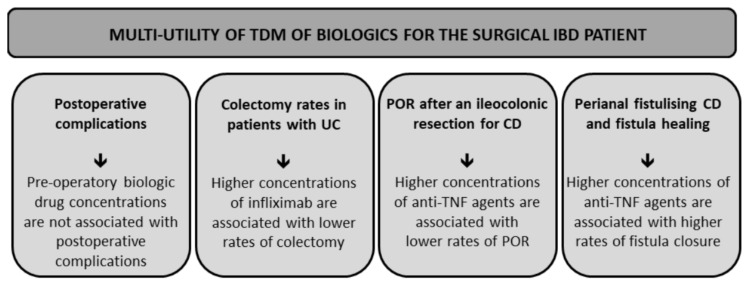
Multi-utility of TDM of biologics for the surgical IBD patient. Legend: UC—ulcerative colitis, POR—post-operative recurrence, CD—Crohn’s disease.

**Table 1 jcm-10-05642-t001:** Therapeutic drug monitoring and perioperative outcome.

Author	Journal (Year)	Type of Study	Number of Patients	Biologic	Outcome Studied	Results
Lau et al. [22]	Ann Surg (2015)	Single-center prospective	123 CD 94 UC	IFX	Overall and infectious postoperative complications with serum concentrations	In CD, IFX concentrations > 3 μg/mL associated with increased overall and infectious complications. IFX concentrations > 8 μg/mL associated with overall complications and readmissions. No relation in UC patients
Fumery et al. [23]	Am J Gastroenterol (2017)	Multicenter prospective	76 CD	IFX/ADA	Overall early (30 day) postoperative complications with serum concentrations	Trough concentrations > 1μg/mL (OR = 0.69, 95%CI: 0.21–2.22) and >3 μg/mL (OR = 0.95, 95%CI: 0.28-2.96) were not associated to increased rates of postoperative complications
Cohen et al. [21]	Gastroenterology (2019) [abstract only]	Multicenter prospective	573 IBD	Anti-TNF agents	Infectious complications and surgical site infections with serum concentrations	No relation between detectable serum drug concentrations or previous exposure to anti-TNF agents with increased infectious postoperative complications or surgical site infections
Parrish AB et al. [25]	Dis Colon Rectum (2021)	Single-center prospective	72 IBD	VDZ	Overall and infectious postoperative complications with serum concentrations	No significant differences in overall postoperative morbidity between detectable (>1.6 μg/ml) and undetectable concentration groups. In CD, there was a significantly lower incidence of postoperative ileus with detectable VDZ concentrations compared to patients with an undetectable VDZ concentration (*p* < 0.04).
Kumar et al. [28]	Gastroenterology (2021) [abstract only]	Single-center prospective	36 IBD	UST	Overall and infectious postoperative complications with serum concentrations	There were no significant differences between the undetectable vs. detectable concentration (≥0.9 μg/ml) groups in regards to overall postoperative morbidity

Legend: CD- Crohn’s disease, UC- ulcerative colitis, IBD- inflammatory bowel disease, IFX- infliximab, ADM- adalimumab, VDZ- vedolizumab, UST- ustekinumab, OR- odds ratio, CI- confidence intervals; TNF- tumor necrosis factor.

**Table 2 jcm-10-05642-t002:** Therapeutic drug monitoring of anti-TNF therapy and postoperative recurrence after ileocolonic resection for CD.

Author	Journal (Year)	Type of Study	Number of Patients	Anti-TNF Agent	Outcome Studied	Results
Regueiro et al. (PREVENT trial) [30]	Gastroenterology (2016)	Multicenter prospective (RCT)	147	IFX	POR rates with serum concentrations (secondary outcome)	Inverse correlation between serum IFX concentrations and POR rates (the higher the concentration, the lower the rates of POR)
Fay et al. [31]	Inflamm Bowel Dis (2017)	Multicenter retrospective	73	IFX/ADA	POR rates with serum concentrations	Lower IFX trough concentrations (median, 1.1 μg/mL versus 2.4 μg/mL; *p* = 0.008) and ATI (5.6% vs. 71.4%, *p* = 0.0001) were significantly associated with endoscopic POR. Same association not observed in ADA patients
Boivineau et al. [33]	J Crohn’s Colitis (2020)	Multicenter prospective	19	ADA	POR rates with serum concentrations	Median serum ADA concentration was 7.95 μg/mL in patients with normal mucosa (Rutgeerts’ score ≤ i1) and 3.25 μg/mL in patients with endoscopic POR (Rutgeerts’ score ≥ i2), respectively (*p* = 0.0485). Serum ADA concentration was inversely correlated to the Rutgeerts’ score (*p* = 0.004)

Legend: IFX—infliximab, ADA—adalimumab, ATI—antibodies toward infliximab, POR—post-operative recurrence, RCT- randomized controlled trial., TNF-tumor necrosis factor.

**Table 3 jcm-10-05642-t003:** Therapeutic drug monitoring of infliximab and colectomy rates in UC.

Author	Journal (Year)	Type of Study	Number of Patients	Anti-TNF Agent	Outcome Studied	Results
Arias et al. [36]	Clin Gastroenterol Hepatol (2014)	Single-center retrospective	112 UC	IFX	Colectomy-free survival with serum concentrations at week 14	A serum IFX concentration > 2.5 μg/mL at week 14 was predictive not only of relapse-free survival (*p* < 0.001), but also of colectomy-free survival (*p* < 0.034).
Papamichael et al. [35]	J Crohn’s Colitis (2016)	Multicenter retrospective	99 UC	IFX	Colectomy rates with serum concentrations at weeks 2 and 6	A ROC analysis identified IFX concentration thresholds of 16.5 and 5.3 μg/mL at weeks 2 and 6, respectively, associated with colectomy. Patients with concentrations on the lower quartile (<10 μg/mL) had higher rates of colectomy at weeks 2 (70%) and 6 (89%), respectively
Battat et al. [41]	Clin Gastroenterol Hepatol (2020)	Single-center retrospective	39 ASUC	IFX	Clearance of IFX with colectomy rates in ASUC	Median baseline calculated clearance of IFX was higher in patients with colectomy at 6 months than in patients without (0.733 vs. 0.569 L/day; *p* = 0.005). A clearance threshold of IFX of 0.627 L/day identified patients who required colectomy (AUC, SN: 80.0%; SP: 82.8%). A higher proportion of patients with IFX clearance of 0.627 L/day or more needed colectomy within 6 months (61.5%) than patients with lower clearance values (7.7%) (*p* = 0.001). Multivariable analysis identified that the baseline IFX clearance value was the only factor associated with colectomy

Legend: IFX—infliximab, UC—ulcerative colitis, ASUC—acute severe ulcerative colitis, ROC—receiver operation curve., TNF-tumor necrosis factor.

**Table 4 jcm-10-05642-t004:** Therapeutic drug monitoring of anti-TNF therapy and perianal fistulizing CD.

Author	Journal (Year)	Number of Patients	Timing of TDM	Anti-TNF Agent	Median (IQR) Drug Concentration in Healed Fistulas, μg/mL	Median (IQR) Drug Concentration in Active Fistulas, μg/mL	Observations
Yarur et al. [44]	Alimentary Pharmacol Ther (2016)	117	Maintenance	IFX	15.8 (9.9–27)	4.4 (0–9.8)	Single-center, cross-sectional retrospective study.
Strik et al. [45]	Scand J Gastroenterol (2019)	IFX = 47 ADA = 19	Maintenance	IFX ADA	6.0 (5.4–6.9) 7.4 (6.5–10.8)	2.3 (1.1–4.0) 4.8 (1.7–6.2)	Single center, cross-sectional retrospective study.
Davidov et al. [46]	J Crohn’s Colitis (2017)	36	Week 2 Week 6 Week 14	IFX	20.0 (16.2–26.3) 13.3 (7.6–19) 4.1 (0.7–5.7)	5.6 (2.8–9.2) 2.6 (0.4–7.0) 0.1 (0.01–2.3)	2-center, retrospective cohort study. Proactive TDM.
Plevris et al. [47]	Eur J Gastroenterol Hepatol (2019)	IFX = 29 ADA = 35	Maintenace	IFX	8.1 12.6	3.2 2.7	Single center, cross-sectional retrospective study.
Papamichael et al. [48]	Am J Gastroenterol (2021)	Induction: 282 Maintenance: 139	Week 14	IFX	9.3 (4.9–16.2)	3.2 (1.1–7.0)	Post hoc analysis of ACCENT II trial. Composite remission (defined as a combined complete fistula response and CRP normalization) data.
Zhu et al. [49]	Dig Dis Sci (2021)	157	6 infusions 12 infusions 18 infusions	IFX	3.5 (0.9–8.7) 2.8 (0.5–6.2) 2.8 (0.4–4.1)	1.9 (0.6–5.2) 1.6 (0.4–3.9) 0.7 (0–2.8)	Retrospective single-center study. Radiological remission data

Legend: IFX—infliximab, ADA—adalimumab, CRP—C reactive protein, TDM—therapeutic drug monitoring, TNF-tumor necrosis factor, IQR: interquartile range.

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
