# Peer review of "Therapeutic Drug Monitoring of Biologics in IBD: Essentials for the Surgical Patient"

_jcm, 2021, doi:10.3390/jcm10235642_

Round 1

Reviewer 1 Report

The authors report a review of papers describing TDM of biologic agents in the setting of surgery for IBD, perioperative and post-surgical, and also with fistulizing disease and in acute severe UC.  The paper is well-written and methodology is generally appropriate.

The paper reports the individual conclusions from each paper but does not particularly synthesize these in a way that is immediately applicable clinically for surgeons and physicians.  The paper would be improved by synthesising the data better.  This could be using graphics or by further modelling analysis. 

The main conclusions of the review are that biologics should be continued perioperatively, and that evidence shows that biologic concentrations should be optimised post-surgery to reduce risk of disease flare.  The wording of the abstract is quite conservative and does not really get these messages across.

One issue that the paper does not discuss is how more severe disease being associated with higher biologic clearance may lead to confounding of some of the study results, ie. it may be the more severe disease rather than lower drug concentrations per se that leads to disease recurrence.  This is helped by prospective studies of which there are not many.  This needs to be discussed.

Specific comments:

Introduction, 3rd paragraph, second sentence - this states that the higher the concentration, the greater chance of induction and maintenance of remission.  I don't think any results support this as worded.  However, the data show that there is a threshold for concentration above which there is increased chance of induction and maintenance of remission.

Part 3. TDM and postoperative recurrence in CD, Second paragraph - there are no statistics for some of the results described. Also in this section there are numbers in brackets without descriptors to know if they are SD, range, 95%CI or other, this needs correction.

Author Response

The authors report a review of papers describing TDM of biologic agents in the setting of surgery for IBD, perioperative and post-surgical, and also with fistulizing disease and in acute severe UC.  The paper is well-written and methodology is generally appropriate.

The paper reports the individual conclusions from each paper but does not particularly synthesize these in a way that is immediately applicable clinically for surgeons and physicians. The paper would be improved by synthesizing the data better.  This could be using graphics or by further modelling analysis.

We have now added Figure 1 to make the concept of the review more comprehensive and immediately applicable clinically for surgeons and physicians. 

The main conclusions of the review are that biologics should be continued perioperatively, and that evidence shows that biologic concentrations should be optimised post-surgery to reduce risk of disease flare.  The wording of the abstract is quite conservative and does not really get these messages across.

Thank you for this observation. We have revised the frame of our abstract in order to highlight the key messages of the review.

One issue that the paper does not discuss is how more severe disease being associated with higher biologic clearance may lead to confounding of some of the study results, ie. it may be the more severe disease rather than lower drug concentrations per se that leads to disease recurrence.  This is helped by prospective studies of which there are not many.  This needs to be discussed.

Thank you for this valuable insight. We have added the following sentence discussing this topic in the post-operative recurrence part of the discussion session.

“Moreover, it should be emphasized that disease recurrence could be significantly influenced by the higher biologic clearance as a consequence of more severe disease at baseline along with the amount of residual inflamed bowel, a confounding factor poorly explored in the available literature.”

Specific comments:

Introduction, 3rd paragraph, second sentence - this states that the higher the concentration, the greater chance of induction and maintenance of remission.  I don't think any results support this as worded.  However, the data show that there is a threshold for concentration above which there is increased chance of induction and maintenance of remission.

Thank you for this observation. We have rephrased as suggested.

Part 3. TDM and postoperative recurrence in CD, Second paragraph - there are no statistics for some of the results described. Also in this section there are numbers in brackets without descriptors to know if they are SD, range, 95%CI or other, this needs correction.

We have corrected as suggested.

Reviewer 2 Report

We congratulate the authors for their excelent review on the applications of TDM in several surgical situations.

Overall the paper is well organized, english is fluent and the most important topics are covered.

Small comments:

1) Most patients who undergo surgery have been previously exposed to biologic therapy [1].
Comment: I have not found any reference to this statement in the article mentioned. Is this based on any study or is it only an author suposition/comment?

2) A serum infliximab concentration greater than 2.5 ug/mL at week 14 was predictive not only of relapse-free survival (p<0.001), but also of colectomy-free survival (p<0.034).
Comment: do you mean P=0.034?

Author Response

We congratulate the authors for their excellent review on the applications of TDM in several surgical situations.

Overall the paper is well organized, English is fluent and the most important topics are covered.

Small comments:

1) Most patients who undergo surgery have been previously exposed to biologic therapy [1].

Comment: I have not found any reference to this statement in the article mentioned. Is this based on any study or is it only an author suposition/comment?

We acknowledge the reviewer for this observation. That was just an author comment and the reference was removed.

2) A serum infliximab concentration greater than 2.5 ug/mL at week 14 was predictive not only of relapse-free survival (p<0.001), but also of colectomy-free survival (p<0.034).

Comment: do you mean P=0.034?

Sorry for this mistake. It is = .034. We have corrected accordingly.